# Observation of space-time surface plasmon polaritons

Naoki Ichiji [1,2], Hibiki Kikuchi[1], Murat Yessenov [3,4], Kenneth L. Schepler [4], Ayman F. Abouraddy [4] ✉ & Atsushi Kubo [5] ✉

Surface plasmon polaritons (SPPs) at metal-dielectric interfaces provide strong out-of-plane confinement enabling nano-scale sensing and imaging, yet diffraction causes spatial delocalization. Conventional strategies to combat diffraction through spatial structuring are inapplicable to dimensionally restricted SPPs, except for nonlocalized cosine plasmons and Airy plasmons that follow curved trajectories. Here we demonstrate space-time SPPs (ST-SPPs), ultrashort (16-fs) diffraction-free SPPs that propagate rectilinearly via precise sculpting of their spatiotemporal spectra. By synthesizing a spatio-temporally structured field in free space and coupling the field to an axially invariant ST-SPP at a metal-dielectric surface, we control the ST-SPP group velocity and propagation characteristics. Time-resolved two-photon fluorescence microscopy reconstructs the surface-bound field in space and time, verifying the predicted spatiotemporal wavefront and diffraction-free propagation. Our work opens new avenues for combining spatiotemporally structured light with the field-localization associated with nanophotonics, and may thus enable novel applications in surface-enhanced sensing and nonlinear optical interactions.

Surface plasmon polaritons (SPPs) are optical surface waves that propagate along metal-dielectric interfaces[1,2]. Because SPPs are strongly localized orthogonally to the interface, they are suitable for a host of photonic applications ranging from biosensing[3] and tweezers[4], to nanofocusing[5,6], modulators[7–9], and harvesting solar energy[10,11], among other fascinating possibilities[12–15]. This strong field confinement of the SPP at the interface is marred by free in-plane propagation, resulting in diffractive spreading of its transverse spatial profile. In addition, SPPs are dispersive, so that the temporal (axial) profile of a pulsed SPP undergoes dispersive spreading (Fig. 1a). Producing diffraction-free, tightly focused, ultrashort pulsed SPPs would yield strong field localization in all dimensions that is maintained over extended propagation distances, which can help enhance many SPP applications.

Axially maintained, in-plane confinement can be provided by surface-patterned waveguide structures[16,17], or by hyperbolic metamaterials that require strongly anisotropic materials[18–21]. In free space, so-called diffraction-free optical beams can maintain their spatial profile axially over considerable distances if their profile conforms to particular functional distributions, e.g., Bessel beams[22]. However, the one-dimensional (1D) transverse nature of SPPs is a restrictive condition with regard to achieving diffraction-free propagation. Indeed, there are no monochromatic diffraction-free light sheets – which are localized along one transverse dimension and uniform along the other – that travel in a straight line[23]. Conventional diffraction-free beams require two transverse dimensions for their viability; so that even a light-sheet that conforms to a Bessel function in 1D is diffractive, in contrast to its 2D counterpart. There are only two exceptions: the cosine wave that is not localized, and the Airy beam that does not travel along a straight line[23]. Both of these monochromatic field structures have been implemented as SPPs: the cosine

[1]Graduate School of Pure and Applied Sciences, University of Tsukuba, Tsukuba-shi, Japan. [2]Institute of Industrial Science, The University of Tokyo, Meguro-Ku, Japan. [3]Harvard John A. Paulson School of Engineering and Applied Sciences, Harvard University, Cambridge, MA, USA. [4]CREOL, The College of Optics & Photonics, University of Central Florida, Orlando, FL, USA. [5]Department of Physics, Institute of Pure and Applied Sciences, University of Tsukuba, Tsukuba-shi, Japan. ✉e-mail: raddy@creol.ucf.edu; kubo.atsushi.ka@u.tsukuba.ac.jp

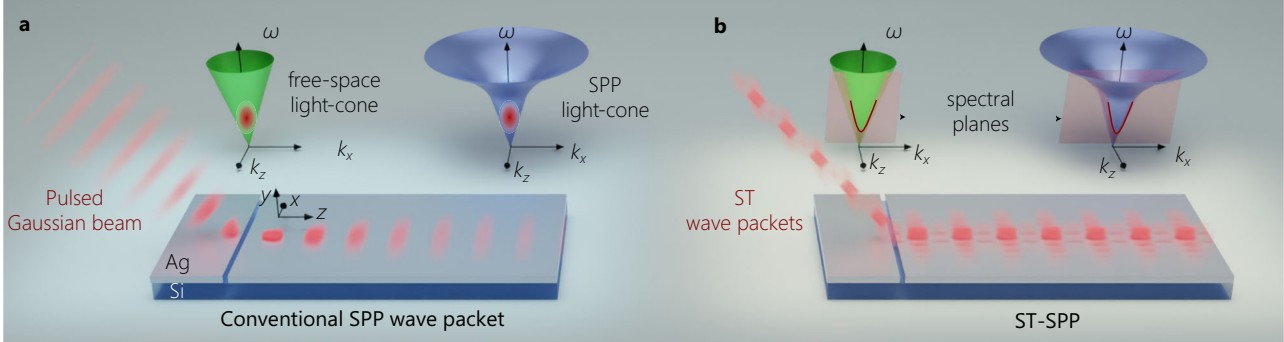

**Fig. 1 | Comparison of SPPs and ST-SPPs at a metal-dielectric interface.**
**a** Schematic of a free-space pulsed beam coupled to a conventional SPP wave packet bound to a metal-dielectric interface via scattering from a nanoslit in the metal surface. The sample is depicted here as a silver (Ag) layer atop a silicon (Si) substrate and clad by free space. The spectral support for the free-space pulsed beam is a 2D domain on the surface of the free-space light-cone $k_x^2 + k_z^2 = (\frac{\omega}{c})^2$, and that for the conventional SPP wave packet is a 2D domain on the surface of the SPP

light-cone $k_x^2 + k_z^2 = k_{SPP}^2$. This conventional SPP wave packet undergoes both spatial diffractive spreading and temporal dispersive spreading. **b** Schematic of a free-space STWP coupled to an ST-SPP on the same sample as in (**a**). The spectral supports for the STWP and for the ST-SPP are curves at the intersection of a tilted spectral plane with the free-space light-cone and the SPP light-cone, respectively. The ST-SPP propagates invariantly on the metal surface without diffraction or dispersion.

plasmon[24,25] and the Airy plasmon[26,27]. Therefore, no diffraction-free SPP traveling along a straight path has been realized to date. This remains a fundamental challenge that cannot be overcome by means of spatially structuring the excitation, which is a consequence of the dimensionality of SPPs as surface waves, along with the monochromaticity of the excitation. Whereas the first restriction cannot be avoided, the second can be readily circumvented by sculpting a pulsed SPP wave packet.

Recently, free-space diffraction-free light sheets that travel along a straight line have been finally realized using a class of pulsed beams known as space-time wave packets (STWPs), which are diffraction-free without imposing a particular 1D transverse spatial profile[28]. Key to this unique feature is lifting the monochromaticity constraint, and then introducing a strict spectral association between the spatial and temporal frequencies undergirding the field spatiotemporal structure, thereby yielding unique characteristics such as propagation invariance in linear media[28–31], tunable group velocities independently of the medium refractive index[30,32], self-healing upon traversing opaque obstacles[33], and anomalous refraction[34]. We have proposed a pulsed SPP beam (or SPP wave packet) that adapts the characteristic structure underpinning freely propagating STWPs to the particular setting of surface waves[35]. Because STWPs are the only propagation-invariant light sheets, they offer the potential for finally yielding diffraction-free SPPs that propagate along a straight line. By restricting the spatiotemporal spectrum of the SPP to the intersection of the SPP light-cone with a tilted spectral plane, one obtains a space-time SPP wave packet (ST-SPP) that inherits many of the unique propagation characteristics of STWPs; in particular, ST-SPPs propagate in a straight line without diffraction or dispersion at a metal-dielectric interface at a group velocity that may be tuned above (superluminal) or below (subluminal) that of a conventional SPP wave packet[35,36]; see Fig. 1b. However, experimentally realizing an ST-SPP and ascertaining its propagation characteristics offers up a host of challenges. Because the typical decay length due to ohmic losses on a metal surface is limited to tens of microns[37–39], unambiguously monitoring the propagation of a pulsed SPP necessitates utilizing a broadband ultrashort pulse of width <20 fs, which then raises the question of coupling a broadband spatiotemporally structured optical field to a surface-bound field[40].

Here we observe for the first time diffraction-free ST-SPPs traveling along a straight path with tunable group velocity at a metal-dielectric interface, which is made possible by resolving the multidimensional experimental challenge outlined above. First, we make use of ultrashort pulses of width 16-fs pulses (110-nm bandwidth),

which correspond to an SPP of axial extent ≈3.6 μm, thereby enabling the monitoring of their propagation over a short distance, and thus enabling us to confirm the tunability of the ST-SPP group velocity above and below that of an SPP as determined by its intrinsic dispersion. Second, we make use of a nano-slit milled into the metal surface for efficient broadband coupling of STWPs from free space to an ST-SPP (predicted theoretically[35,41] and demonstrated with minimally structured fields[42]), while maintaining the structure of the target spatiotemporal spectrum. Third, the use of ultrashort pulses plays a crucial role in the phase-sensitive detection of the spatially and temporally resolved surface-bound field through two-photon fluorescence excited by a reference pulse interfering with the evanescent tail of the SPP at the metal surface. We thus reconstruct the surface-bound field for a variety of ST-SPPs, in addition to conventional SPP wave packets, and verify the details of their spatiotemporal structure, including a predicted signature tilt in their phase front with respect to the propagation direction. Furthermore, by exciting conventional SPP wave packets and ST-SPPs of the same bandwidth, we confirm the diffractive spreading of the former and the diffraction-free propagation of the latter. These results establish a new form of SPP wave packets that offers unprecedented control over their propagation characteristics. This may enable novel applications in enhanced surface nonlinear optical interactions mediated by the tunable group velocity of ST-SPPs, in generating radiation through accelerating ST-SPPs[43], and launching isolated propagating topological SPP spin textures[13,44,45] or SPPs endowed with transverse orbital angular momentum[46,47].

## Results
### Theoretical formulation
The conceptual formulation of ST-SPPs is best understood by visualizing their spectral representation on the surface of the light-cone. For (2 + 1)D pulsed optical fields restricted to one transverse dimension $x$ in addition to the axial dimension $z$, the light-cone is the geometric representation of the dispersion relationship involving the transverse wave number $k_x$ (or spatial frequency), the longitudinal wave number $k_z$, and the temporal frequency $\omega$. In free space, the light-cone is $k_x^2 + k_z^2 = (\frac{\omega}{c})^2$, and that for an SPP at a metal-dielectric interface is $k_x^2 + k_z^2 = k_{SPP}^2(\omega)$; where $c$ is the speed of light in vacuum, $k_{SPP} = \frac{\omega}{c}\sqrt{\frac{\epsilon_m \epsilon_d}{\epsilon_m + \epsilon_d}}$, and $\epsilon_m$ and $\epsilon_d$ are the relative permittivities of the metal and dielectric, respectively (Fig. 1a). In free space, where a (2 + 1) D pulsed beam (or wave packet) takes the form of a light-sheet that is

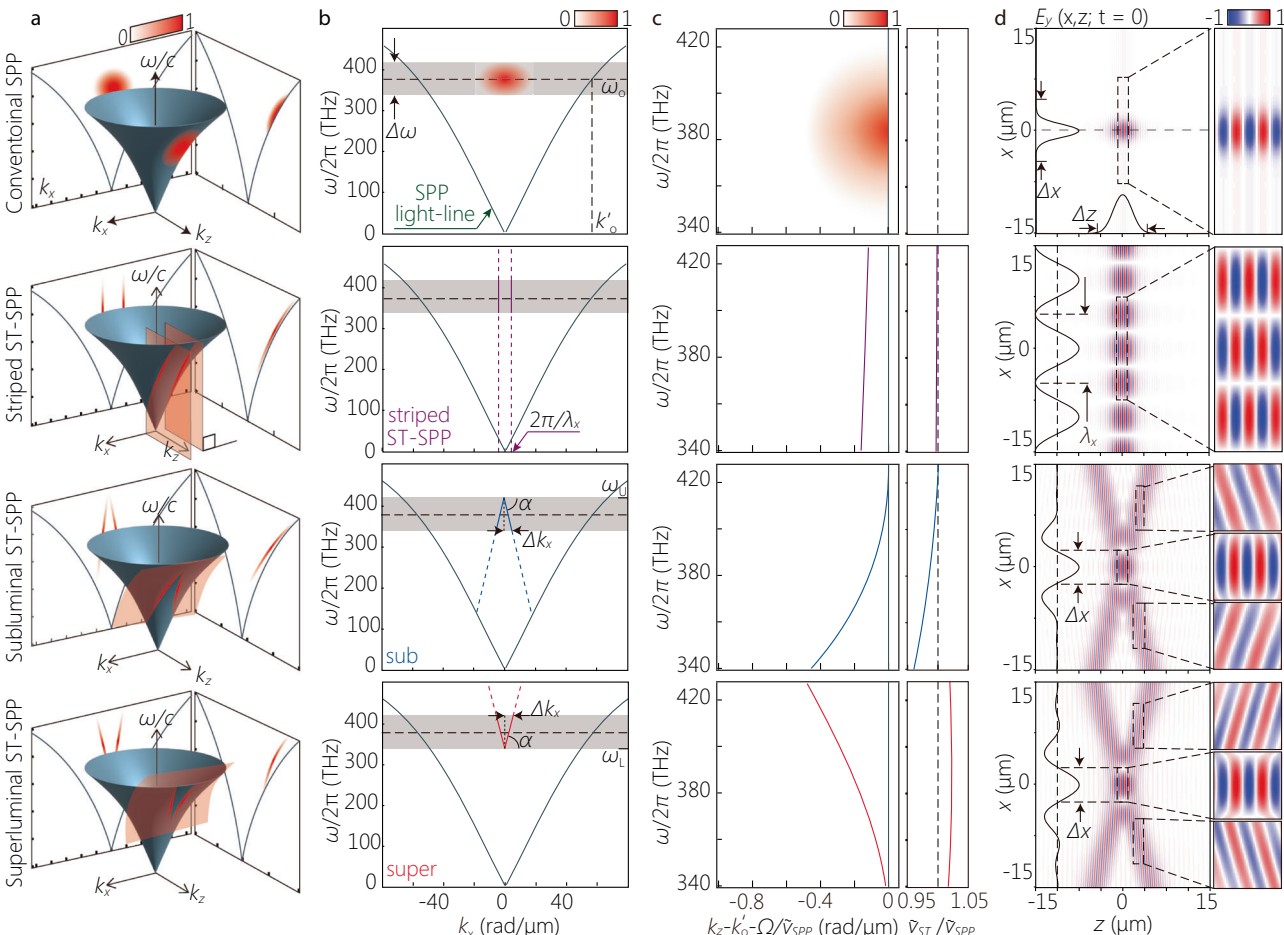

**Fig. 2 | Spectral representation of SPPs and ST-SPPs.** The first row corresponds to a conventional SPP wave packet with $\Delta x = 9\,\mu m$; the second to a striped ST-SPP with $\lambda_x = 10\,\mu m$; the third to a subluminal ST-SPP with $\Delta x = 5\,\mu m$; and the fourth to a superluminal ST-SPP with $\Delta x = 5\,\mu m$. (**a**) The spectral support on the surface of the SPP light-cone $k_x^2 + k_z^2 = k_{SPP}^2$ in $(k_x, k_z, \frac{\omega}{c})$-space (shown in red). **b** The projection of the spectral support in **a** onto the $(k_x, \frac{\omega}{c})$-plane. This spectral projection is invariant upon coupling the field from free space to the metal-dielectric interface (as a consequence of conservation of energy and transverse momentum). The gray band corresponds to the bandwidth $\Delta\omega$ of the laser pulse used in our experiments, and the dashed horizontal line is at $\frac{\omega_o}{2\pi} \approx 375$ THz ($\lambda_o \approx 800$ nm in free space). **c** The

spectral projection onto the $(k_z, \frac{\omega}{c})$-plane after implementing $k_z \rightarrow k_z - k_o' - \frac{\Omega}{v_{SPP}}$ for clarity, where $\Omega = \omega - \omega_o$. The panel to the right shows the group velocity of the wave packet $\tilde{v} = \frac{d\omega}{dk_z}$ normalized to $\tilde{v}_{SPP}$. The spectral range on the vertical axis corresponds to the bandwidth of the laser pulse. **d** The calculated out-of-plane real part of the field distribution $E_y(x, y, z; t)$ at $y = 0$ and $t = 0$ (Methods). The black curves on the left and bottom are cross-sections at $z = 0$ and $x = 0$, respectively. The right panels show magnified views of the dashed squares in the left panel, highlighting the structure of the phase fronts.

uniform along $y$ and localized along $x$, its spectral support is a two-dimensional (2D) region on the surface of the free-space light-cone (Fig. 1a). Such a light-sheet undergoes diffractive spreading with free propagation. A conventional SPP wave packet is a $(2 + 1)$D surface wave with finite transverse-spatial extent along $x$ in addition to surface localization along $y$, whose spectral support in turn corresponds to a 2D region on the surface of the SPP light-cone (Fig. 1a)[35,48]. This conventional SPP wave packet undergoes both diffractive spreading in space and dispersive spreading in time.

Ideal STWPs are pulsed beams that propagate rigidly in linear media without diffraction or dispersion, and whose propagation characteristics can be tuned largely independently of the material parameters[23,28]. Rather than the 2D spectral support on the free-space light-cone associated with conventional pulsed light sheets, the spatiotemporal spectrum of an STWP is restricted to a 1D curve (a conic section) at the intersection of the light-cone with a plane that is parallel to the $k_x$-axis and makes an angle $\theta$ with the $k_z$-axis (Fig. 1b)[28]. This plane, $\omega - \omega_o = (k_z - k_o)c \tan\theta$, results in a straight-line spectral projection onto the $(k_z, \frac{\omega}{c})$-plane, which indicates a fixed group velocity $\tilde{v} = c \tan\theta$ and the absence of dispersion to all orders[49]; here $\omega_o$ is a

fixed carrier frequency, and $k_o = \frac{\omega_o}{c}$ is its associated wave number. Moreover, the one-to-one association between $\omega$ and $|k_x|$ guarantees diffraction-free propagation of the time-averaged intensity[23]. Similarly for ST-SPPs, their spectral support as shown in Fig. 1b is also a 1D curve at the intersection of the SPP light-cone with the plane $\omega - \omega_o = (k_z - k_o')\tilde{v}_{SPP} \tan\theta$, where $\tilde{v}_{SPP}$ is the group velocity of a plane-wave pulsed SPP at $\omega = \omega_o$, and $k_o'$ is the SPP wave number at $\omega_o$[35]. The spectral projection onto the $(k_z, \frac{\omega}{c})$-plane for the ST-SPP is also a straight line, indicating that – in addition to being diffraction-free – ST-SPPs propagate dispersion-free at a group velocity $\tilde{v} = \tilde{v}_{SPP} \tan\theta$ despite the intrinsic GVD associated with freely propagating SPPs (Fig. 1b)[35].

We show in Fig. 2 (first row) the spectral support for a conventional SPP wave packet on the SPP light-cone (Fig. 2a), its spectral projections onto the $(k_x, \frac{\omega}{c})$ and $(k_z, \frac{\omega}{c})$ planes (Fig. 2b, c), along with the out-of-plane field profile (Fig. 2d). We assume a separable Gaussian spectrum in space and time, and take into account the temporal bandwidth $\Delta\lambda \approx 110$ nm (FWHM) utilized in our experiments (Methods). Crucially, the phase front for this conventional SPP wave packet is orthogonal to the propagation axis $z$ (Fig. 2d).

Producing an ideal ST-SPP with a bandwidth $\Delta\lambda = 110$ nm is prohibitive, as can be understood by considering a so-called 'striped ST-SPP' whose spatial profile results simply from the interference of a pair of plane waves with fixed spatial frequencies $\pm k_x$ (i.e., the field takes the transverse form $\cos k_x x$)[42]. Consequently, a striped ST-SPP is separable with respect to space and time, and it thus does not offer the possibility of tuning the group velocity (indeed, its group velocity is $\tilde{v} \approx \tilde{v}_{SPP}$; Methods). We plot in Fig. 2a–c (second row) the spectral support for a striped ST-SPP, which is the intersection of iso-$|k_x|$ planes with the SPP light-cone. The out-of-plane field is separable, extended and periodic along $x$ (with period $\lambda_x \equiv 2\pi/k_x = 10\,\mu m$), and its phase front is orthogonal to the $z$-axis, leading to a lattice-like spatial distribution at any instant $t$ (Fig. 2d).

A broadband ST-SPP is composed of a multiplicity of cosine SPPs with different $k_x$, each of which is associated with a prescribed $\omega$, such that $k_z$ is maintained in a linear relationship with $\omega$. However, maintaining this linear relationship between $k_z$ and $\omega$ over a large temporal bandwidth $\Delta\omega$ is prohibitive for SPPs because the curvature of the SPP light-line dictates a large accompanying spatial bandwidth $\Delta k_x$. Indeed, even if $\tilde{v}$ it remains within 1% of $\tilde{v}_{SPP}$, we still need $\frac{\Delta k_x}{k_o} \approx 0.23$ (with the parameters used in our experiments below) to maintain the spatiotemporal spectrum of an ideal ST-SPP over the large bandwidth $\Delta\lambda \approx 110$ nm used here. Because our experimental scheme conserves the spatial bandwidth $\Delta k_x$ from free space to the surface-bound field, a large numerical aperture is required. We define $k_x(\lambda) = \frac{2\pi}{\lambda}\sin\{\varphi(\lambda)\} = \frac{2\pi}{\lambda_x}$ in free space, where $\lambda$ is the free-space wavelength, $\varphi(\lambda)$ is the its propagation angle with the $z$-axis, and $\lambda_x = \lambda/\sin\varphi(\lambda)$ is the transverse period of the field along $x$. Our system is limited to a maximum angular acceptance $\varphi_{max} \approx \pm 5°$, a numerical aperture (NA) of $\approx 0.09$, which restricts the minimum spatial feature size to $\approx 9\,\mu m$ which falls considerably short of the requirement for an ideal ST-SPP over $\Delta\lambda = 110$ nm. To address this challenge, we have implemented a compromise regarding the structure of the realized ST-SPP with respect to an ideal ST-SPP: (1) we maintain the one-to-one correspondence between $|k_x|$ and $\omega$ that is necessary for diffraction-free propagation; (2) we maintain control over the group velocity $\tilde{v}$ of the ST-SPP away from that for a conventional SPP $\tilde{v}_{SPP}$ in both the subluminal ($\tilde{v} < \tilde{v}_{SPP}$) and superluminal ($\tilde{v} > \tilde{v}_{SPP}$) regimes; however (3) we allow the spectral projection onto the $(k_z, \frac{\omega}{c})$-plane to be curved such that it lies in the $(k_z, \frac{\omega}{c})$-plane between the SPP light-line and the linear spectral projection for an ideal ST-SPP (Fig. 2c). Consequently, we can operate with the available system NA, but at the cost of vouchsafing the complete elimination of GVD. This is a minimal disadvantage in light of the short SPP propagation distance limited by ohmic losses.

The spectral support of the realized ST-SPP is still a 1D curve on the surface of the SPP light-cone, but this curve results from the intersection of the SPP light-cone with a curved planar surface (rather than a plane) designed to yield a V-shaped spectral projection onto the $(k_x, \frac{\omega}{c})$-plane[50]. For a subluminal ST-SPP ($\tilde{v} < \tilde{v}_{SPP}$), this projection takes the form $\omega - \omega_U = c|k_x|\tan\alpha$, where $\omega_U = \omega_o + \frac{\Delta\omega}{2}$ is the maximum frequency, $\omega < \omega_U$, $\alpha$ is the angle with the $k_x$-axis, and $\tan\alpha < 0$. For a superluminal ST-SPP ($\tilde{v} > \tilde{v}_{SPP}$), we have $\omega - \omega_L = c|k_x|\tan\alpha$, where $\omega_L = \omega_o - \frac{\Delta\omega}{2}$ is the minimum frequency, $\omega > \omega_L$, and $\tan\alpha > 0$. In both cases, $\omega_o$ is the central frequency, so that $\omega_o - \frac{\Delta\omega}{2} < \omega < \omega_o + \frac{\Delta\omega}{2}$. This curved surface remains parallel to the $k_x$-axis, but its projection onto the $(k_z, \frac{\omega}{c})$-plane is no longer a straight line and is instead a curve that remains closer to the SPP light-line than in the case of the ideal ST-SPP (to remain within the system NA). The analytical relationship between ST-SPP group velocity $\tilde{v}$ and the angle $\alpha$ is derived in Methods. The GVD coefficient for a conventional SPP wave packet in this case is $2.5 \times 10^3$ fs²/mm, and is $2.5 \times 10^3$, $2.3 \times 10^2$, and $2.0 \times 10^2$ fs²/mm for a striped ST-SPP (with $\lambda_x = 10\,\mu m$), a subluminal ST-SPP (with $\alpha = -64.5°$), and a superluminal ST-SPP (with $\alpha = 64.5°$), respectively.

We plot in Fig. 2 the spectral representation for the subluminal ST-SPP (third row) and the superluminal ST-SPP (fourth row). Of particular interest is the spatial profile of the out-of-plane field for both ST-SPPs, which is no longer separable as a result of the non-separability of their spatiotemporal spectra. Instead, we observe an X-shaped profile reminiscent of free-space STWPs[49,51]. Crucially, the direction of the phase fronts varies across the wave front. In the vicinity of the center $x = 0$, the phase front is orthogonal to the propagation axis. However, along the branches of the X-shaped profile, the phase fronts are tilted with respect to the $z$-axis. Crucially, the sign of this phase-front tilt switches between the subluminal and superluminal regimes, and the magnitude of the tilt angle increases with the deviation of $\tilde{v}$ from $\tilde{v}_{SPP}$[52].

## Experimental arrangement

We sketch in Fig. 3a the optical arrangement for synthesizing free-space STWPs, launching them into ST-SPPs on a metal-dielectric interface via scattering from a nano-slit, and observing the propagation of surface-bound ST-SPPs via spatially and temporally resolved two-photon fluorescence produced from the interference of the ST-SPP with a free-space reference pulse. We start with pulses from a Ti:sapphire laser oscillator of bandwidth $\Delta\lambda \approx 110$ nm (FWHM), center wavelength $\lambda_o \approx 800$ nm, and a 10-fs transform-limited pulse duration (FWHM). However, the measured pulse width at the sample surface is $\Delta T \approx 16$ fs (FWHM) because of residual chirp in the optical system.

**Synthesis of ST-SPPs.** The STWPs are synthesized from the pulsed laser via spatiotemporal spectral phase modulation as established in refs. 28,49. The pulse spectrum is spatially resolved by a grating (300 lines/mm), the first diffraction order is selected and collimated using a cylindrical lens, which then impinges on a reflective, phase-only spatial light modulator (SLM); see Fig. 3a. This SLM imparts a 2D phase distribution to the incident wave front to deflect each wavelength $\lambda$ by angles $\pm\varphi(\lambda)$ with respect to the $z$-axis, where $\sin\{\varphi(\lambda)\} = \pm(1 - \frac{\lambda}{\lambda_c})\cot\alpha$. Here $\lambda_c$ is the minimum ($\lambda > \lambda_c$) or the maximum ($\lambda < \lambda_c$) wavelength in the superluminal or subluminal regimes, respectively. The reflected phase-modulated wavefront is directed to a second grating identical to the first, whereupon the pulse is reconstituted to produce the STWP. For simplicity, the synthesis system in Fig. 3a is depicted as retro-reflecting from the SLM (whereupon the second grating coincides with the first); see Methods for details.

**Sample preparation and launching ST-SPPs onto a metal-dielectric interface.** The sample is a 100-nm-thick Ag film deposited on a silicon substrate. A crucial experimental challenge is to couple a broadband free-space STWP to a surface-bound ST-SPP at the metal-dielectric interface. We have recently examined theoretically an alternative approach to launching free-space STWPs into ST-SPPs that relies on scattering from a nano-slit in the metal surface (Fig. 3b)[41]. This coupling strategy retains a high efficiency over the bandwidth used here, and is thus preferable to conventional approaches for SPP coupling (e.g., gratings or evanescent prism-coupling) that modify the transverse wave number or do not operate over large bandwidths. Such a nano-slit conserves the transverse wave number $k_x^{free} = k_x^{SPP} = k_x$, where $k_x^{free}$ and $k_x^{SPP}$ are the transverse wave numbers for the free-space field and the launched SPP on either side of the nano-slit, respectively, so that the axial wave number is $k_z = \pm\sqrt{k_{SPP}^2 - k_x^2}$.

Two sets of 170-nm-wide, 100-nm-deep nano-slits were milled into the Ag film (Fig. 3c, e). One set of nano-slits has a lateral length of 200 $\mu m$ and is used to launch ST-SPPs onto the Ag surface when illuminated with a free-space STWP (Fig. 3c). When illuminated with a conventional pulsed beam of large transverse extent along $x$, a conventional SPP wave packet is launched with a transverse width equal to that of the

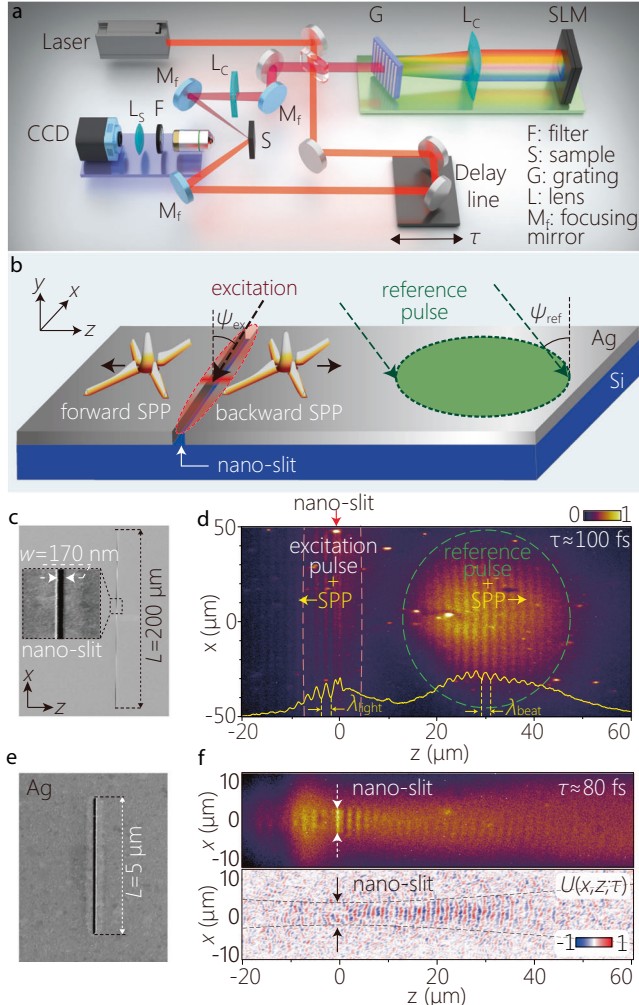

**Fig. 3 | Experimental arrangement for synthesizing, launching, and observing ST-SPPs. a** Schematic of the setup for synthesizing free-space STWPs, launching them onto the sample, and imaging of the surface-bound field. **b** Schematic of the sample irradiation configuration. The STWP incidence angle is $\psi_{ex} = -45°$ to the surface normal, and the reference pulse incidence angle is $\psi_{ref} = 45°$. **c** SEM micrograph of the nano-slit milled into the Ag film. The inset shows a section of the nano-slit. **d** Two-photon fluorescence beat profile $I(x, z; \tau)$ at a fixed delay $\tau$ for a conventional SPP wave packet launched from the nano-slit. The yellow curve at the bottom is the axial profile $I(z; \tau) = \int dx\, I(x, z; \tau)$ at a fixed $\tau$. The red arrow at the top indicates the nano-slit location. Dashed orange lines enclose the irradiation area of the excitation, and the dashed green circle encloses that for the reference pulse. **e** SEM micrograph of the 5-$\mu$m-long nano-slit used to launch a conventional SPP wave packet of that width. **f** The two-photon fluorescence beat pattern (upper panel) after irradiating the nano-slit in (**e**), and the extracted field $U(x, z; \tau)$ (lower panel), both at fixed delay $\tau \approx 80$ fs (see Supplementary Movies 1 and 2). The dotted black curves correspond to the width of a Gaussian beam of width 5 $\mu$m (1/$e^2$ full width of the intensity), and serve as a guide for the eye.

incident field (Fig. 3d). The second set of nano-slits has a reduced lateral length of 5 $\mu$m (Fig. 3e), and is used to launch conventional SPPs with a 5-$\mu$m-wide spatial profile (Fig. 3f). We have experimentally confirmed the efficacy of the nano-slit coupling methodology with striped ST-SPPs[42], the spatiotemporally separable building blocks of ST-SPPs that comprise a single spatial frequency. Our results here confirm the theoretical predictions in ref. 41 regarding the broadband coupling of STWPs to ST-SPPs. Finally, the sample is coated by a 30-nm-thick dye-doped PMMA film to form a two-photon fluorescent layer.

**Spatiotemporal characterization of the ST-SPPs.** A portion of the initial laser pulse (20% by power) is split off to serve as a reference pulse, while the remainder (80%) is directed to the above-described STWP synthesis system (Fig. 3a). The synthesized STWP that excites the ST-SPP is incident obliquely at an angle $\psi_{ex} = -45°$ with respect to the sample normal, and is focused via a combination of spherical mirror and cylindrical lens onto the nano-slit, which launches it onto the sample surface (Fig. 3b; Methods)[42]. The incident free-space field is launched by the nano-slit in both the forward and backward directions[53,54] (Fig. 3b). The reference pulse traverses an optical delay line $\tau$ and is then focused onto the metal surface away from the nano-slit to interact with the surface-bound SPP. The interference beat pattern of the SPP and the reference pulse excites two-photon fluorescence from the polymer layer[55]. Because we aim at detecting exclusively the interference of the reference pulse with the SPP, after eliminating any interference between the incident STWP excitation and the excited ST-SPP, we make use of the backward-coupled SPP ($z > 0$) rather than the stronger forward-coupled SPP ($z < 0$); see Fig. 3b.

## Observation of SPPs and ST-SPPs
**Time-resolved measurements.** We show in Fig. 3d the fluorescence profile of a conventional SPP wave packet with a large transverse spatial profile excited by irradiating the nano-slit in Fig. 3c with a conventional pump pulse after setting the SLM phase to 0 everywhere; i.e., $\varphi(\lambda) = 0$. The beat profile to the left of the nano-slit ($z < 0$) results from self-interference of the launched SPP and the incident free-space excitation, which is thus stationary and independent of $\tau$ because the reference pulse is not involved. The two-photon fluorescence beat profile to the right is delimited by the irradiation area of the reference pulse on the sample surface (green dashed circle; focal spot $30 \times 60\ \mu m^2$). The delay $\tau$ is adjusted to correspond to the propagating SPP wave packet reaching $z \sim 30\ \mu$m. We define $t$ as the time incurred by the surface-bound SPP (that travels at a group velocity $\tilde{v}$) to propagate from the nano-slit to the observation location. The time $t$ will be different from the free-space delay $\tau$ placed in the path of the focused reference pulse (that travels at a group velocity $c / \sin \psi_{ref}$ along the metal surface) to reach the same location and produce the interference beat pattern; see Methods for the conversion between the delay $\tau$ and the time $t$.

We are now in a position to compare the time-resolved measurements of conventional SPP wave packets and ST-SPPs. We first investigate a conventional SPP wave packet where the initial laser pulses are focused onto the metal surface at the location of a 5-$\mu$m-long nano-slit (Fig. 3e). A conventional SPP wave packet is thus launched at the metal surface with a flat intensity profile, 5-$\mu$m transverse spatial width, and 16-fs temporal pulse width. The recorded two-photon fluorescence beat profile resulting from the launched SPP interfering with the reference pulse (focused spot size $60 \times 15\ \mu m^2$) is shown in Fig. 3f. The intensity drops due to ohmic losses and diffractive spreading, and only a faint arc-shaped beat pattern is visible when the SPP wave packet reaches $z \sim 40\ \mu$m. From this intensity profile we extract the SPP field distribution $U(x, z; \tau)$ plotted in the bottom panel of Fig. 3 f at fixed delay $\tau \approx 80$ fs (Methods); see Supplementary Movies 1 and 2.

In Fig. 4 we plot the experimentally recorded and the computed two-photon fluorescence beat profiles at a fixed delay $\tau$ (Methods) for a conventional SPP of width 5 $\mu$m (Fig. 4a,b); a striped ST-SPP with transverse period $\approx 10\ \mu$m (Fig. 4c,d); a subluminal ST-SPP with $\alpha = -64.5°$ (Fig. 4e,f); and a superluminal ST-SPP with $\alpha = 64.5°$ (Fig. 4g, h). The differences in the wave packet center position and the extent of the interference envelopes are attributed to uncertainties in the time origin, spatial inhomogeneity in the fluorescent film, and variations in the intensity distribution of the probe beam. Typical time-resolved measurements of ST-SPPs are provided in Supplementary Movies 3–6. Computed spatiotemporal profiles capture all the key features of their

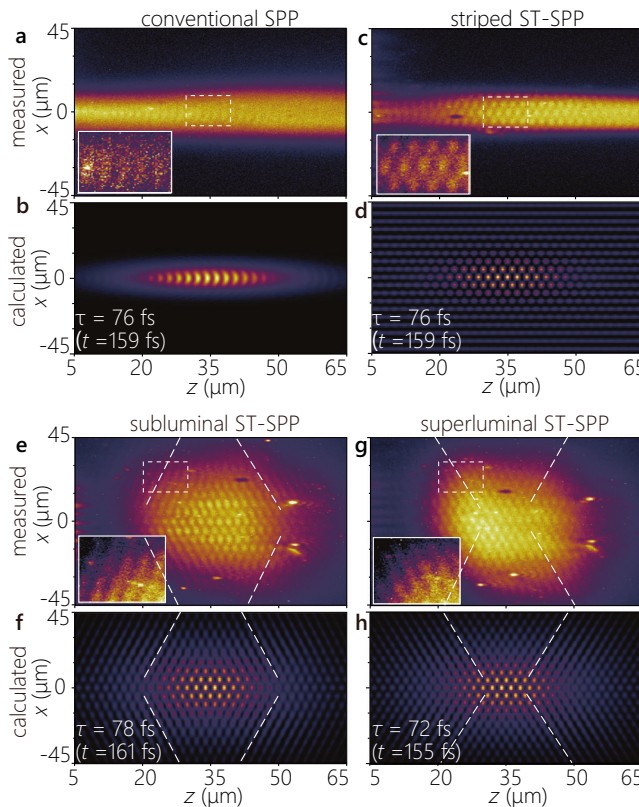

**Fig. 4 | Time-resolved two-photon fluorescence beat profiles for SPPs and ST-SPPs. a** Measured two-photon fluorescence beat profiles $I(x, z; \tau)$ at a fixed delay $\tau$ produced by the interference of a conventional SPP wave packet (of transverse spatial width 5 $\mu$m) with the reference pulse. Inset is a contrast-adjusted magnified view of the area enclosed in the dashed white rectangle ($10 \times 16 \, \mu m^2$) to highlight the flat wave front. **b** Calculated profile $I(x, z; \tau)$ corresponding to the measurement in (**a**); see Methods. **c**, **d** Same as **a**, **b** but for a striped ST-SPP with $\lambda_x \approx 10 \, \mu$m. **e**, **f** Same as **a**, **b** but for a subluminal ST-SPP; and **g**, **h** for a superluminal ST-SPP. The white dashed lines are guides for the eye representing the propagating wavefronts, and the insets highlight the tilted wave fronts at the edges of the ST-SPPs. All the two-photon fluorescence micrographs are acquired at a delay $\tau \approx 75$ fs. The time $t$ listed at the bottom left of the calculated profiles is that needed for each SPP to reach $z = 35 \, \mu$m (Methods). The focused spot size of the reference pulse was $\approx 60 \times 15 \, \mu m^2$ in (**a**–**d**), and $\approx 30 \times 50 \, \mu m^2$ in (**e**–**h**).

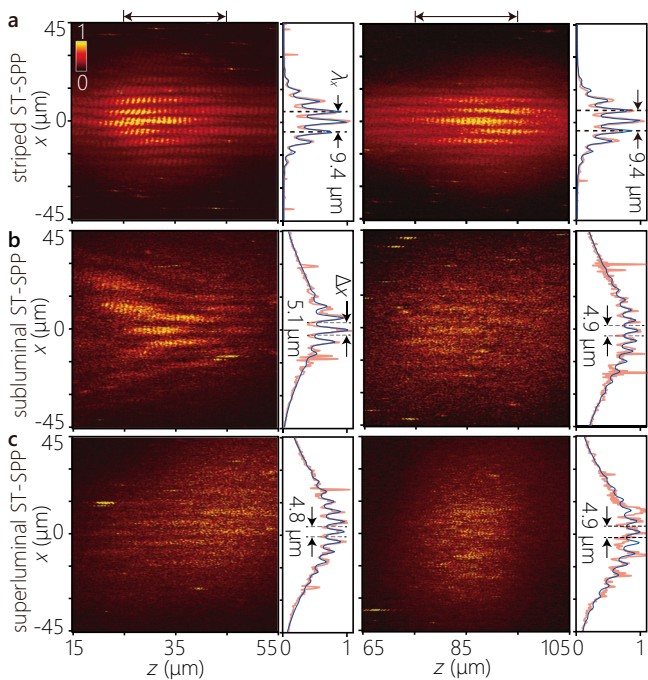

**Fig. 5 | Diffraction-free propagation of ST-SPPs.** We plot the measured time-averaged intensity profiles for **a** a striped ST-SPP, **b** a subluminal ST-SPP, and **c** a superluminal ST-SPP. Time-averaged intensity profiles $I(x, z; \tau)$ are measured at two locations for the reference pulse. The left panel corresponds to the reference pulse focused near the nano-slit at $15 < z < 55 \, \mu$m (with delay in the range $\tau \approx 20 - 130$ fs), and the right panel to the reference pulse focused away from it at $65 < z < 105 \, \mu$m ($\tau \approx 130$–$230$ fs). The intensity profiles $I(x)$ plotted to the right of each panel are obtained by axial integration $I(x) = \int dz \, I(x, z)$ over the range identified by the black arrows on top (red curves), along with least-square fits (blue curves). The overall intensity distribution of the fluorescence profiles is affected by the irradiated region of the reference light. The Rayleigh ranges for the corresponding beam waists are 18.2 $\mu$m for the striped ST-SPP and 21 $\mu$m for the superluminal and subluminal ST-SPPs, respectively.

**Diffraction-free propagation of ST-SPPs.** To verify the diffraction-free behavior of the ST-SPPs, we evaluated the intensity profiles $I(x, z) = \int d\tau \, |U_{\text{beat}}(x, z; \tau)|^2$ at different axial positions by time-integration of the extracted field profiles $U_{\text{beat}}(x, z; t)$ obtained by removing background intensity from the two-photon fluorescence beat profiles (Methods). We plot in Fig. 5 the time-averaged intensity profiles for the striped ST-SPP (Fig. 5a), the subluminal ST-SPP (Fig. 5b), and the superluminal ST-SPP (Fig. 5c). For each case, we plot the intensity profiles obtained after focusing the reference pulse at two different axial positions along the sample surface: at $z \approx 35 \, \mu$m and $z \approx 85 \, \mu$m measured from the location of the nano-slit.

In contrast to a conventional SPP wave packet whose profile diffracts rapidly with propagation (Fig. 3f), the profiles of the ST-SPPs remain unchanged over the same propagation distance. In the case of the striped ST-SPP, this is to be expected because it contains only a single transverse wave number (i.e., an extended cosine wave profile). The limited transverse profile is a result of the narrow reference pulse irradiation spot utilized at the sample surface. The intensity profiles in Fig. 5a demonstrate that the striped ST-SPP maintains the same profile over a range of $\approx 5 \times$ the Rayleigh range ($z_R \approx 18.2 \, \mu$m; Methods). The transverse profiles of the subluminal (Fig. 5b) and superluminal (Fig. 5c) ST-SPPs are also maintained over the distance examined, which is over $\approx 4 \times$ Rayleigh lengths (the Rayleigh length is $z_R \approx 21 \, \mu$m). The reduced visibility in these time-averaged profiles compared to the time-resolved measurements in Fig. 4e,g is attributed to the integration over the full spatiotemporal field, which includes a broadband

measured counterparts. Crucially, the different phase front structures predicted in Fig. 2d are clearly observable, as depicted in the insets of the corresponding panels in Fig. 4. Whereas the conventional SPP has a flat phase front orthogonal to the $z$-axis (Fig. 4a, b, and Fig. 2d, first row), the striped ST-SPP has a checkered lattice-like structure (Fig. 4c,d, and Fig. 2d, second row). The phase front for the subluminal ST-SPP away from its center is tilted with respect to the $z$-axis (Fig. 4e, f, and Fig. 2d, third row). The sign of this phase-front tilt is reversed for the superluminal ST-SPP (Fig. 4g, h, and Fig. 2d, fourth row). The remaining quantitative differences between the computed and measured profiles in Fig. 4 are attributed to the residual chirp in the laser pulses, additional chirp encumbered in the STWP synthesis system, and spectral dissipation at the metal surface.

We have thus confirmed that the nano-slit is capable of launching free-space STWPs onto the metal surface as ST-SPPs, and that our experimental configuration can capture the time-resolved structure of the surface-bound, propagating ST-SPPs. We proceed to utilize this capability to verify two key features of ST-SPPs: their diffraction-free propagation, and the controllability of their group velocity ($\tilde{v}$) above and below that of a conventional SPP ($\tilde{v}_{\text{SPP}}$).

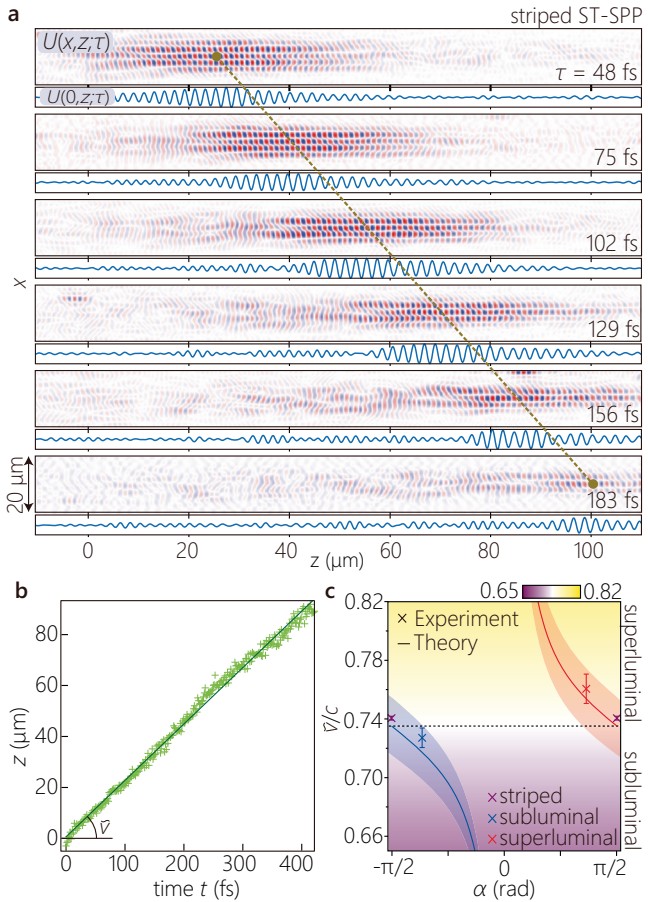

**Fig. 6 | Tuning the group velocity of ST-SPPs. a** The real part of the extracted field profiles $U(x, z; \tau)$ for a striped ST-SPP at different delays $\tau$ (listed on the right). We plot underneath each profile the longitudinal section $U(x = 0, z; \tau)$. Each peak position was obtained by least-square fitting with a sinusoidally modulated Gaussian wave packet. **b** The axial center coordinate $z$ of the peak of the intensity beat-profile from **a** plotted as a function of real time $t$ (of the propagating striped ST-SPP) calculated from the external delay time $\tau$ (of the reference pulse; Methods). The corresponding range of the values of the delay is $\tau = 0 - 200$ fs. **c** Measured and calculated group velocities for the ST-SPPs (striped, superluminal, and subluminal). The solid curves are the calculated group velocities for ST-SPPs with $\Delta x = 5 \, \mu$m. The uncertainty bands surrounding the theoretical curves $\tilde{v}$ result from a $\pm 2.5$-nm uncertainty in the thickness of the 30-nm-thick fluorescent PMMA layer.

pedestal that partially obscures the localized structures.[28,49] In all cases, the integrated cross-section along $x$ plotted in the right panel is reasonably fitted by the (superposition) of Gaussian functions (representing the focused spot of the reference pulse) and the square of the sinusoidal-Gaussian function (representing the ST-SPP). The propagation distance here is limited by the ohmic losses. It is expected that the diffraction-free behavior extends significantly beyond $4z_R$, which can become manifest by increasing the NA and reducing the width $\Delta x$ of the ST-SPP (reducing $z_R$ significantly below the loss-restricted propagation distance).

**Tuning of the group velocity of ST-SPPs.** A unique feature of ST-SPPs is the possibility of tuning their group velocity by modifying only the structure of their spatiotemporal spectrum – without changing the sample itself. The group velocities of the ST-SPPs were estimated by examining a sequence of time-resolved two-photon fluorescence beat profiles. All group velocity measurements were performed at a fixed location on the sample to eliminate variability arising from spatial inhomogeneity in the fluorescent film.

Additionally, the delay time $\tau$ was adjusted such that the center of the ST-SPP wave packet appeared at $z = 0$ when $t = 0$. An example is shown in Fig. 6a corresponding to the striped ST-SPP. For different values of the relative delay $\tau$, we extract the field structure $U(x, z; \tau)$ and then fit (in the least-square sense) a sinusoidally modulated Gaussian function to the axial field $U(0, z; \tau)$ at each $\tau$ (Fig. 6a). We take the peak of this fitted function to be the axial center-of-weight $z$ of the propagating ST-SPP. The delay $\tau$ was converted into propagation time $t$ of the ST-SPP on the metal surface by factoring in that the surface-bound ST-SPP travels at a group velocity $\tilde{v}$, whereas the reference pulse travels at $c / \sin \psi_{\text{ref}}$ along the surface (Methods). We plot in Fig. 6b the measured axial positions $z$ of the striped ST-SPP with real time $t$, and the estimated slope yields the group velocity $\tilde{v}_{\text{SPP}} = (2.22 \pm 0.004) \times 10^8$ m/s $(0.74c)$, which is consistent with the group velocity of a conventional SPP $\tilde{v}_{\text{SPP}} = 2.22 \times 10^8$ m/s obtained from the first derivative of the dispersion curve of the SPP mode at the PMMA-coated Ag surface.

We repeated the procedure for the subluminal and superluminal ST-SPPs, and the estimated group velocities are $\tilde{v}_- = (2.18 \pm 0.02) \times 10^8$ m/s $(\approx 0.73c)$ and $\tilde{v}_+ = (2.28 \pm 0.03) \times 10^8$ m/s $(\approx 0.76c)$, respectively. The relative relationships of the measured group velocities $\tilde{v}_- < \tilde{v}_{\text{SPP}} < \tilde{v}_+$ are thus consistent with theoretical expectations and with the wavefront tilts in the two-photon fluorescence beat patterns in Fig. 4. We plot in Fig. 6c the measured group velocities compared to $\tilde{v}$ calculated as a function of the opening angle $\alpha$ after taking into consideration the uncertainty resulting from the finite accuracy of determining the thickness ($30 \pm 2.5$ nm) of the fluorescent PMMA layer.

## Discussion

We have confirmed that broadband ST-SPPs with complex separable and non-separable spatiotemporal structures can be excited at a metal-dielectric interface from a free-space STWP via a nano-slit milled into the metal surface. By appropriately sculpting the free-space spatiotemporal spectrum, we synthesized a variety of ST-SPP structures, monitored the time-resolved surface-bound fields, observed their signature wave-front tilt, confirmed their diffraction-free propagation along a straight line, and verified that their group velocity $\tilde{v}$ can be tuned above (superluminal) and below (subluminal) that of a conventional SPP wave packet having the same bandwidth. The demonstrated tuning range of $\tilde{v}$ was limited by the numerical aperture of our optical system (NA $\approx 0.09$), given the broad spectral bandwidth (~110 nm) required to ensure sufficient temporal resolution for resolving the propagation dynamics. However, the coupling efficiency is not significantly affected by the group velocity, and simulations have confirmed that ST-SPPs with a broader group velocity range can be excited[41]. Accordingly, broader tunability of $\tilde{v}$ can be achieved by increasing the NA or reducing the spectral bandwidth.

This work builds upon our previous demonstration of striped ST-SPPs[42], sharing the femtosecond laser source and a single-nano-slit light-SPP coupling strategy. While retaining these foundations, the present study significantly extends the experimental framework to access a broader class of ST-SPPs[35]. Specifically, whereas the prior work treated separable ST-SPPs associated with a single transverse spatial frequency $k_x$, the current setup accommodates spatiotemporal spectra with finite spatial bandwidths, thereby enabling the excitation of non-separable ST-SPPs with tunable group velocity.

By incorporating an interferometric detection scheme, we can now obtain time-resolved measurements of the evolving ST-SPP field–rather than the time-averaged intensity profiles used previously. This capability allows direct evaluation of both the spatiotemporal structure of the wave packet and its propagation characteristics. The detection scheme developed here may be useful investigating the temporal evolution of light-matter interaction of

structured light on small lengths and time scales. By demonstrating the feasibility of launching broadband ST-SPPs at a metal-dielectric interface, this proof-of-principle experiment thus extends the frontier of the burgeoning field of spatiotemporally structured optical fields into plasmonics and surface waves in general, including potentially water waves and surface acoustic waves.

A plethora of avenues for further investigation can now be pursued. The experimental arrangement is so far limited by the numerical aperture within the paraxial regime. An immediate extension would involve increasing the numerical aperture of the free-space synthesis system, thereby increasing the spatial bandwidth of the STWP, resulting in stronger focusing of the excited ST-SPPs, with wavelength-scale and potentially sub-wavelength scale transverse spatial width[35]. Here, the unique advantages of ST-SPPs with regards to diffraction-free propagation become more pronounced, potentially reaching $\sim300 \times z_R$ for $\Delta x \approx 3\,\mu m$ (reaching 1/e of its initial on-axis peak value at $z = 2100z_R$)[35]. Another improvement in field localization can be achieved in the unique context of SPPs. The central spatial feature of the ST-SPP rides atop a broad pedestal, which drops significantly as we increase the spectral uncertainty undergirding the launched free-space STWP[23]. In previous work in free space, efforts have been devoted to reducing the spectral uncertainty in order to increase the propagation distance $L_{max}$. This is not a pressing concern here for ST-SPPs (as it is for free-space optical communications, for example) in light of the typical SPP decay lengths[35,52]. As long as $L_{max} \sim 40\,\mu m$, one may increase $\delta\lambda$ accordingly and thus substantially reduce the background term in the time-averaged intensity[23].

Recent studies have shown that structured SPPs can possess both in-plane and out-of-plane spin angular momentum[56,57], which raises the intriguing possibility of structured SPP wave packets, such as ST-SPPs having three-dimensional spin textures[52], and hence the potential for observing the interaction between structured SPPs and electron spin in magnetic materials. Another intriguing possibility is to exploit axial acceleration of ST-SPPs[58] in contrast to transverse acceleration of SPPs[43] induced by curved Airy beams is envisioned as a potential route for inducing radiation from the metal surface. Finally, our work opens new avenues for combining spatiotemporally structured light with the field localization associated with nanophotonics. The tunable group velocity of the ST-SPPs (which also includes a negative-$\tilde{v}$ regime[35]) may enable phase-matching of nonlinear effects in thin films. Finally, ST-SPPs can be designed to become omni-resonant[59] with surface Fabry-Pérot cavities[60], such that the full bandwidth of broadband ST-SPPs couples to a single axial resonant mode. This raises the possibility of resonant – yet broadband – field enhancement at the metal interface for surface-enhanced Raman scattering (SERS) detection of molecules and pathogens.

## Methods
### Calculating the SPP and ST-SPP electric-field distributions
The calculated profiles of the out-of-plane electric fields for the SPPs and ST-SPPs plotted in Fig. 2d assume a 100-nm-thick Ag film on a Si substrate, with a 30-nm-thick PMMA coating clad with free space. The SPP dispersion curve $k_{SPP}(\omega)$ was calculated using a model proposed by Pockrand[61], and employed the dielectric function for Ag proposed by Rakic et al.[62]. The measured refractive index of PMMA is $n \approx 1.53$ at the operating wavelength. We take the pulse spectrum to be $\tilde{E}(\omega)$, which is the Fourier transform of the pulse profile $E(t) = \int d\omega \tilde{E}(\omega)e^{-i\omega t}$. We assume a Gaussian profile in time:

$$E(t) = E_o\, e^{-i\omega_o t} \exp\left(-\frac{t^2}{22(\Delta T)^2}\right), \qquad (1)$$

where $\Delta T = 10$ fs is the FWHM of the intensity profile, and $\frac{\omega_o}{2\pi} = 375$ THz.

In general, the out-of-plane field is given by:

$$E(x,z;t) = \iint dk_x\, d\omega\, \tilde{\psi}(k_x,\omega)\, e^{i(k_x x + k_z z - i\omega t)}, \qquad (2)$$

where $\tilde{\psi}(k_x,\omega)$ is the spatiotemporal spectrum, which is the Fourier transform of the initial field $E(x,0;t)$. The first row in Fig. 2 corresponds to a conventional SPP wave packet, where $\tilde{\psi}(k_x,\omega)$ is separable, $\tilde{\psi}(k_x,\omega) = \tilde{\psi}(k_x)\tilde{E}(\omega)$; here $\tilde{\psi}(k_x)$ is the spatial spectrum of the excited SPP and assumed also to have a Gaussian profile.

The second row in Fig. 2 corresponds to a striped ST-SPP. The spatiotemporal spectrum is once again separable $\tilde{\psi}(k_x,\omega) = \{\delta(k_x - k_{xo}) + \delta(k_x + k_{xo})\}\tilde{E}(\omega)$. The electric-field distribution is thus given by:

$$E(x,z;t) \propto \cos(k_{xo}x) \int d\omega\, \tilde{E}(\omega)e^{i(k_z(\omega)z - \omega t)}, \qquad (3)$$

where the spatial frequency $k_{xo}$ is fixed and is independent of $\omega$, so that the transverse spatial DoF is separable with respect to the axial spatial and temporal DoFs. Consequently, the striped ST-SPP is diffraction-free (the transverse spatial profile does not change with propagation), but it is not localized.

The third and fourth rows in Fig. 2 correspond to subluminal and superluminal ST-SPPs, respectively. The spatiotemporal spectrum is no longer separable, in which $\omega$ and $|k_x|$ are in one-to-one correspondence after selecting a value for the angle $\alpha$. The field distribution for the ST-SPP is given by:

$$E(x,z;t) \propto \int d\omega\, \tilde{E}(\omega) \cos\{k_x(\omega)x\} e^{i\{k_z(\omega)z - \omega t\}}, \qquad (4)$$

which is not separable with respect to the spatial and temporal DoFs. In all cases we have $k_z(\omega) = \sqrt{k_{SPP}^2(\omega) - k_x^2}$, and we take the same spectral profile $\tilde{E}(\omega)$ used above for the conventional SPP, the striped ST-SPPs, and the ST-SPPs.

### Relationship between the spectral tilt angle $\theta$ and the opening angle $\alpha$
We define the axial group velocity $\tilde{v}$ of the ST-SPP in the usual way as $\tilde{v} = \left(\frac{dk_z}{d\omega}\big|_{\omega_o}\right)^{-1}$. For the V-shaped spectra realized here (Fig. 2a, b, third and fourth rows), tuning the opening angle $\alpha$ varies $\theta$, and thus in turn $\tilde{v}$. For subluminal ST-SPPs we have $\omega - \omega_U = c|k_x|/\tan\alpha$, where $\omega_U = \omega_o + \frac{\Delta\omega}{2}$ is the upper temporal frequency in the spectrum, $\omega < \omega_U$ and $\tan\alpha < 0$ ($-90° < \alpha < 0$). The axial wave number is $k_z(\omega) = \sqrt{k_{SPP}^2(\omega) - k_x^2(\omega)}$, and we make use of a Taylor expansion for $k_{SPP}(\omega)$: $k_{SPP} \approx k_o' + \frac{\Omega}{v_{SPP}} + \frac{1}{2}k_2\Omega^2$, where $\tilde{v}_{SPP}$ and $k_2$ are the group velocity and GVD coefficient for a plane-wave pulsed SPP, respectively, and $k_x^2(\omega) = \left(\frac{\Omega - \Delta\omega/2}{c\tan\alpha}\right)^2$. We can thus write $k_z = \sqrt{A + 2B\Omega + C\Omega^2}$, where: $A = k_o'^2 - \frac{(\Delta k)^2}{4\tan^2\alpha}$, $B = \frac{k_o'}{v_{SPP}} + \frac{\Delta k}{2c\tan^2\alpha}$, $C = k_o'k_2 + \frac{1}{v_{SPP}^2} - \frac{1}{c^2\tan^2\alpha}$, and $\Delta k = \frac{\Delta\omega}{c}$. We now have the following expression for the group velocity:

$$\tilde{v} = \left(\frac{dk_z}{d\Omega}\big|_{\Omega=0}\right)^{-1} = \frac{\sqrt{A}}{B}. \qquad (5)$$

Defining a group index $\tilde{n} = \frac{c}{v} = \cot\theta$ for the ST-SPP and $\tilde{n}_{SPP} = \frac{c}{v_{SPP}}$ for a conventional SPP wave packet, we have:

$$\tilde{n} = \left(\tilde{n}_{SPP} + \frac{\Delta k/k_o'}{2\tan^2\alpha}\right)\left(1 + \frac{(\Delta k/k_o')^2}{8\tan^2\alpha}\right), \qquad (6)$$

so that $\tilde{n} > \tilde{n}_{SPP}$ ($\tilde{v} < \tilde{v}_{SPP}$) as expected for a subluminal ST-SPP.

For the superluminal ST-SPP we have $\omega - \omega_L = c|k_x|\tan\alpha$, where $\omega_L = \omega_o - \frac{\Delta\omega}{2}$ is the lower temporal frequency in the spectrum, $\omega > \omega_L$ and $\tan\alpha > 0$ ($0 < \alpha < 90°$). The coefficients $A$ and $C$ are identical to those for the subluminal ST-SPP, and $B = \frac{k_o}{v_{SPP}} - \frac{\Delta k}{2c\tan^2\alpha}$, so that the group index for the ST-SPP is:

$$\tilde{n} = \left(\tilde{n}_{SPP} - \frac{\Delta k/k_o'}{2\tan^2\alpha}\right)\left(1 + \frac{(\Delta k/k_o')^2}{8\tan^2\alpha}\right), \qquad (7)$$

and to first order we have $\tilde{n} \approx \left(\tilde{n}_{SPP} - \frac{\Delta k/k_o'}{2\tan^2\alpha}\right)$, $\tilde{n} < \tilde{n}_{SPP}$ ($\tilde{v} > \tilde{v}_{SPP}$) as expected for a superluminal ST-SPP.

### Synthesizing free-space STWPs

The light source used in our experiments is a custom-built Ti:sapphire laser oscillator with a transform-limited pulse duration of 10 fs, center wavelength $\lambda_o = 800$ nm, FWHM-bandwidth $\Delta\lambda \approx 110$ nm, a repetition rate 90 MHz, and average power $\approx 400$ mW. The spectrum is measured via a spectrometer (Ocean Optics HR4000) after coupling to an optical fiber with diameter 200 $\mu$m. Because of residual chirp accumulated through various optical components such as lenses, gratings, and the SLM, the pulses are not transform-limited, and instead have a pulse duration of $\Delta T \approx 16$ fs at the sample surface. The pulse width is estimated by a home-built auto-corrrelator[55].

A portion of the laser power (20%) is reserved for subsequent use as a reference pulse. The remainder of the laser power (80%) is directed by a beam splitter to an optical system for synthesizing STWPs built along the lines described in ref. 28. The spectrum of the femtosecond pulse is spatially resolved by a grating (300 lines/mm), collimated by a cylindrical lens $L_c$ (focal length $f = 250$ mm), and directed to a 2D phase-only, reflective SLM (Hamamatsu X13138-07). Each wavelength $\lambda$ occupies a column of the SLM, along which the SLM imparts a spatial phase distribution $\pm\frac{2\pi}{\lambda}\sin\{\varphi(\lambda)\}x$. The overall 2D phase distribution imparted to the wave front corresponds to each wavelength being deflected by a prescribed propagation angle $\pm\varphi(\lambda)$ to yield the target relationship between $k_x$ and $\lambda$. The reflected phase-modulated, spectrally resolved wavefront is directed to a second grating (identical to the first), whereupon the pulse is reconstituted to produce an STWP, which is then directed to the sample to excite an ST-SPP.

### Sample fabrication

The sample was prepared by first depositing a 100-nm-thick Ag layer on a Si substrate via sputtering. The nanoslit structure was then milled into the Ag surface using a focused ion beam. The dimensions of the nanoslit are as follows: width $w = 170$ nm, depth $h = 100$ nm into the Ag layer, and extending across the sample by a length $L = 200$ $\mu$m (Fig. 3c). These were used to excite the ST-SPPs, including the striped ST-SPPs, and the subluminal and superluminal ST-SPPs (Figs. 4, 6, and 5). This nano-slit was also used to excite the conventional plane-wave SPP wave packet (Fig. 3d). A second set of nano-slits was prepared with the same dimensions $w$ and $h$, but with length $L = 5$ $\mu$m (Fig. 1e). These nano-slits were used to excite conventional SPP wave packets with 5-$\mu$m-wide transverse profiles (Fig. 3f).

The full sample area was spin-coated with a 30-nm-thick poly (methyl methacrylate) (PMMA) layer doped with a laser dye (coumarin 343). The thickness of the coating was measured by spectral ellipsometry (Uvisel plus), from which we estimated an uncertainty in the

coating thickness of $\sim$ 5 nm. This in turn results in an uncertainty in estimating the SPP group velocity of $\sim 0.1 \times 10^8$ m/s.

### SPP detection

Two wave packets overlap and interfere in the fluorescent polymer layer: the free-space incident reference pulse and the surface-bound SPP. The interference of these two wave packets produces a beat profile that excites two-photon fluorescence. The two-photon-fluorescence emission from the sample surface is collected with an objective lens (M Plan Apo SL20X, Mitutoyo) equipped with a band-pass filter transmitting light in the range 475–495 nm (Semrock, FT-02-485/20-25), followed by a CCD camera (Rolera EM-C2, QImaging). The two-photon fluorescence signal is expected at a wavelength $\approx 490$ nm from the dye used.

In this study, we chose a field of view of approximately $150 \times 150$ $\mu$m, which allowed simultaneous imaging of both the light-SPP coupler and the full extent of the ST-SPP wave packet, while maintaining the spatial resolution required for accurate phase retrieval. As a result, the observation range was primarily limited to approximately 100 $\mu$m from the slit, despite the ST-SPPs propagating beyond this distance.

### Calculating the SPP intensity profile

The intensity distribution $I(x, z; \tau)$ on the sample surface detected by the CCD camera is the time-average of the fourth power of the total electric field[55]:

$$I(x, z; \tau) = \int dt\, \left(|E_{SPP}(x, z; t) + E_{free}(x, z; t + \tau)|^2\right)^2; \qquad (8)$$

Here, a proportionality constant is ignored because we are only interested in the relative intensity. The term $E_{SPP}$ and $E_{free}$ are the scalar field components of the excited SPP and the free-space fields, respectively, that overlap in the PMMA film. The interference of the two fields produces a spatial pattern having a spatial beat wavelength $\lambda_{beat}$ given by:

$$\lambda_{beat} = \frac{2\pi}{|k_z - k_o \sin\psi_{ref}|}, \qquad (9)$$

where $k_z(\omega_o, k_x)$ is the SPP axial wave number at $\omega = \omega_o$, and $k_o \sin\psi_{ref}$ is the in-plane axial wave number of the incident free-space field of the reference pulse. Similarly, the interference beat between the pump-probe pulse pair is $\lambda_{light} = \frac{2\pi}{|k_o \sin\psi_{ex} - k_o \sin\psi_{ref}|}$, where $k_o \sin\psi_{ex}$ is the in-plane axial wave number of the incident free-space field of the pump pulse. Therefore, the intensity, phase, and spatial distribution of the SPP wave packet can be extracted from the observed beat pattern through appropriate analysis. However, it is important to note that the beat pattern is not a direct representation of the SPP field itself. In particular, due to the interference-based nature of the detection method, the spatial frequency of the beat does not directly correspond to the SPP wavevector, and the detected signal decays more rapidly than the actual SPP field because it scales with the fourth power of the electric field amplitude. When employing $\psi_{ex}$ and $\psi_{ref}$ with the same signs to observe forward-coupling SPP, $\lambda_{light} \approx \lambda_{beat}$ if the values of $\psi_{ex}$ and $\psi_{ref}$ are comparable, making differentiation between them difficult. Consequently, we employed a contrasting configuration in which the signs of $\psi_{ex}$ and $\psi_{ref}$ are opposite of each other, such that $\lambda_{light}$ is significantly less than $\lambda_{SPP}$, thereby facilitating their delineation. The interference beats caused by this pump-probe pulse pair can be seen as the intensity distribution moving from right to left at the beginning of Supplementary Movies 1 and 3.

Intensity profiles in Fig. 4 were calculated from the two-photon fluorescence signal resulting from the interference of the ST-SPP and the incident free-space conventional reference pulse. The intensity

profile was calculated using Eq. (8) and assuming that both the ST-SPP and reference pulses are of width ≈ 16 fs as determined experimentally.

## Removing the background

To extract the field distribution $U(x, z; \tau)$ from the measured two-photon fluorescence, we need to experimentally estimate the background constants. This is achieved by changing the delay $\tau$ in the path of the reference pulse by 2.7 fs, which corresponds to moving the excitation-reference interference beat profile by the laser carrier wavelength laser $\lambda_o$ (a phase shift of $2\pi$ rad), and carrying out the time-resolved measurements. To obtain the background intensity, we first obtain the time-resolved field $U(x, z; \tau)$ from the interference beat profiles, and then obtain their average $\bar{U}(x, z; \tau)$ evaluated over an optical cycle:

$$\bar{U}(x, z; \tau) = \frac{1}{m} \sum_{i=1}^{m} U\left(x, z; \tau + (i-2) \times \frac{2.7}{m}\right). \quad (10)$$

In our work here, we have used $m = 2$ or $4$ according to the exposure time used to avoid bleaching the fluorescence layer during laser irradiation. This averaging eliminates the excitation-reference oscillatory beat profile and extracts only the delay-independent background. This procedure assumes that minimal spatial change occurs in the SPP profile while advancing $\tau$ by 2.7 fs. By subtracting the averaged profile $\bar{U}(x, z; \tau)$ from the time-resolved profiles $U(x, z; \tau)$ we obtain the background-free profile of the pump-probe beat pattern at $\tau$, $U_{beat}(x, z; \tau) = U(x, z; \tau) - \bar{U}(x, z; \tau)$. This procedure yields the field profile displayed in the bottom panel of Figs. 3f, 6a, and Supplementary Movies 2 and 4.

## Rayleigh range

The intensity distribution of the microscopic fluorescence beat images obtained via two-photon fluorescence are not proportional to the electric field intensity. We estimate the beam waist of the striped ST-SPP by obtaining $\lambda_x$ by fitting the striped ST-SPP to a sinusoidal distribution along $x$. For a striped ST-SPP with $\lambda_x \approx 9.4\,\mu m$, the beam waist is $2.15\,\mu m$, corresponding to a Rayleigh range $z_R \approx 18.2\,\mu m$ at a wavelength of 800 nm.

## Group-velocity calculation

The two-photon fluorescence beat pattern is formed by the spatial overlap of the SPP propagating at a group velocity $\tilde{v}$ and the reference pulse traveling along the sample surface at a group velocity $\tilde{v} = c/\sin\psi_{ref}$, where $\psi_{ref}$ is the reference-pulse incident angle. We need to convert the external relative delay $\tau$ between the excitation and reference pulses to the propagation time $t$ of the surface-bound SPP. As the SPP travels after a time $t$ a distance $z_1 = \tilde{v}t$, whereas the delayed reference pulse travels a distance $z_2 = \frac{c}{\sin\psi_{ref}}(t - \tau)$. For the two wave packets to meet at $z_1 = z_2 = z_o$, and thus a two-photon fluorescence beat profile to be observed in the vicinity of $z_o$, we have $\tau = z_o\left(\frac{1}{\tilde{v}} - \frac{\sin\psi_{ref}}{c}\right)$. This leads to the sought-after conversion $t = \tau(1 - \frac{\tilde{v}}{c}\sin\psi_{ref})$. This conversion between $t$ and $\tau$ was used in plotting Fig. 4 and 6b.

## Data availability

All data supporting the findings of this study are available within the article and its Supplementary Movies. The raw fluorescent images are openly accessible on Figshare under the DOI: 10.6084/m9.figshare.30225220.

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

## Acknowledgements

N.I. was supported by JSPS KAKENHI (JP20J21825, JP23KJ0355, JP25K17906). A.K. was supported by JSPS KAKENHI (JP16823280, JP22H05131, JP23H04575, JP25H01606, JP25K01648) and MEXT Q-LEAP ATTO (JPMXS0118068681). M.Y., K.L.S., and A.F.A. were supported by the U.S. Office of Naval Research (ONR) under contract N00014-20-1-2789; M.Y. and A.F.A. were also supported under contract N00014-17-1-2458. The sample fabrication was supported by the Nanotechnology Platform Project (JPMXP09F-17-NM-0068); and by the Nanofabrication Platform of NIMS and University of Tsukuba.

## Author contributions

A. F. A. and A. K. developed the concept and supervised the research; N. I. designed the experiments, carried out the measurements, and analyzed the data, with assistance from H. K; M. Y. and K. L. S. advised on the experiments and analyzed the data; all authors contributed to writing the paper.

## Competing interests

The authors declare no competing interests.

## Additional information

**Peer review information** : *Nature Communications* thanks the anonymous reviewers for their contribution to the peer review of this work. A peer review file is available.

