## [Transparent Peer Review file · Nature Communications]

Observation of space-time surface plasmon polaritons

Corresponding Author: Dr Atsushi Kubo

Version 0:

Reviewer comments:

Reviewer #1

(Remarks to the Author)

This work reports on the synthesis and detection of ST-SPPs in different coupling regimes. The unique spatiotemporal structure of these waves allows them to travel long distances without diffraction. This work builds on the authors' previous reports of STWPs and striped ST-SPPs (their refs. 28 and 42). The main innovation here is the synthesis of ST-SPPs which may be tuned into striped, subluminal, and superluminal regimes, as they show experimentally. The present results and the detailed analysis and modeling provided are noteworthy, but it would be helpful to more clearly draw a comparison between the previous work (ref. 42) and this report since similar theory, methods, and experimental configuration are used to generate the striped ST-SPPs. A few other comments below.

Since the reference pulse co-propagates with the SPP as shown in Fig. 3d, I would expect the beat and self interference frequencies to be different but they appear very close in the figure. Can the authors comment on this?

The ST-SPP images recorded near the trench (Fig. 4) agree with the simulated field profiles. For verifying diffraction free propagation, it is not clear why 85 μm was chosen as the appropriate distance. There is a trade-off between dissipation and intensity, but farther away from the trench would make this result more convincing since the propagation length for a NIR SPP in Ag is at least twice that distance, in theory. A second point is that the profiles shown in Fig. 5 seem to be limited by the spot size of the reference beam. As such, if the ST-SPP diffracts beyond this area the signal will fade and give the appearance of diffraction free propagation. The concept of diffraction free propagation is supported by the theory provided, however, it is unclear to what extent the spatial convolution with the reference beam influences the Fig. 5 profiles.

The ST-SPPs possess complex phase structure which could be exploited or may make some applications more difficult. Can the authors comment on how the phase profiles will influence some of the nanophotonic applications they propose such as surface-enhanced sensing?

Other comments:

- Consider moving the last sentence of the Fig. 2 caption to the beginning of the caption to help guide the reader.
- In Fig. 3b, the red excitation arrow is difficult to see since it aligns with the trench.
- The text on page 10 refers to Fig. 3g but there is no such label for the field profile $U(x,z;t)$.
- Consider making the labels c, d, g, and h bold in Fig. 4 like the others.

Reviewer #3

(Remarks to the Author)

The authors of the manuscript "observation of space-time surface plasmon polaritons" have experimentally prepared a non-diffracting space-time surface plasmon polariton (ST-SPP) wavepacket by coupling a precisely tailored optical field having a restricted 1D curvature in the spectral representation. Using two-photon fluorescence microscopy equipped with interferometry technique, they have unambiguously imaged the propagation of such ST-SPP on silver surfaces over tens of microns and hundreds of femtoseconds and reconstructed its spatiotemporal field. Finally, the capability of tuning, despite the small range, of the group velocity of the ST-SPP wavepacket was shown. I find the result to be of interest for the plasmon community, and the manuscript is well prepared. I therefore suggest the authors to address the following points before I can recommend it for publication.

Q1. Since the excitation light for SPP fields needs spatiotemporal modulation of the wavefront, it unecessarily inherent some spatiotemporal chirp of the excitation pulse. Could the authors discuss a bit of the effect of chirping, so that what would be the ultimate limit for the spatial confinement in the lateral dimension (k_z in Fig. 1) of the ST-SPP. Will the angle of excitation matter in generation of the striped ST-SPP?

Q2. The striped ST-SPP in Fig. 4c seem to have interference features extending upto 65 microns, much longer than that of the conventional SPPs in Fig. 4a, although they are measured a the same time, and their respective simulations have similar spatial distribution. So what is the cause of this "potential" elongation of the interference contrast?

Q3. Related to Q1, the lateral confinement (now becomes x-axis, however), is not much different in Fig. 4a and c. So what is the advantage of having striped ST-SPP in this case? I assue by having ST-SPP, one already reduces the intensity of the excitation light, and the coupling efficiency, possibly.

Q4. The superluminal and subluminal propagation seem to have unsatisfying contrast in the experiments. Can the authors comment in a bit more detail why this occurs? This is rather critical in a sense that whether further tuning of the group velocity is feasible for ST-SPP. For example, only limited tuning range was experimentally demonstrated, desipte the thermoetical curve has a broader capability of tuning. If coupling efficiency reduces significantly for SPPs of different velocities, then it kind of loses the advantange as an information carrier.

Version 1:

Reviewer comments:

Reviewer #1

(Remarks to the Author)

The revised manuscript and comments received by the authors address all of my questions and concerns. I enthusiastically support its publication at this time.

Reviewer #3

(Remarks to the Author)

The authors have largely revised the manuscript based on my comments and concerns, which addresses most of the issues. The reported technique is indeed a good alternative for imaging nanoscale fields, and certainly deserve further investigation and broad interest. I think the manuscript is now in good shape for publication.

Responses to Reviewer 1

Reviewer's comment

This work reports on the synthesis and detection of ST-SPPs in different coupling regimes. The unique spatiotemporal structure of these waves allows them to travel long distances without diffraction. This work builds on the authors' previous reports of STWPSs and striped ST-SPPs (their refs. 28 and 42). The main innovation here is the synthesis of ST-SPPs which may be tuned into striped, subluminal, and superluminal regimes, as they show experimentally. The present results and the detailed analysis and modeling provided are noteworthy,

Authors' response:

We appreciate the reviewer for the positive evaluation of our manuscript.

but it would be helpful to more clearly draw a comparison between the previous work (ref 42) and this report since similar theory, methods, and experimental configuration are used to generate the striped ST-SPPs. A few other comments below.

Authors' response:

We thank reviewer for raising this issue. Ref. [42] is positioned as a preliminary demonstration of a limited case of ST-SPPs within the scope of this work. Accordingly, this work and Ref. [42] share certain attributes, while introducing several significant differences. Below we outline the common attributes and differences between the previous work [42] and this work.

Common attributes between this work and previous work [42]:

- Both works exploit the same femtosecond laser source.
- Same coupling technique via nano-slit is used to excite the ST-SPP in this work and in the previous work.

Novelty of this work:

(1) Fundamental differences in target / theory

- The theory in this work is a generic for ST-SPP with controllable group velocity and finite spatial bandwidth, whereas in ref. [42] it was simplified to the case of single k_x .
- The ST-SPP here is non-separable in space and time, whereas in ref. [42] it is separable.
- The group velocity is tunable across superluminal and subluminal regimes.

(2) Advances in experimental setup

- Direct observation of the spatiotemporal evolution of the ST-SPP wave packet during propagation was enabled by incorporating a pump-probe interferometer-based imaging system.
- Newly designed custom phase masks were implemented on the SLM.
- A silver film was employed instead of gold, enhancing the SPP propagation length.

(3) New observations enabled by these improvements

- The full spatiotemporal evolution of ST-SPPs was captured far beyond the slit and into the far field, whereas Ref. [42] observed only the region immediately adjacent to the slit and only at the moment just after excitation.
- Group velocity tunability across superluminal and subluminal regimes was demonstrated, consistent with the generalized theory.
- We demonstrated that the diffraction-free behavior of ST-SPPs is maintained over a distance of 100 μm , exceeding the typical propagation length.

- Transverse spatial localization, which was absent in ref. 42, is demonstrated in this work through the generation of ST-SPPs with finite spatial bandwidth, resulting in three-dimensional localization.

We have also included the differences in the Discussion section of the revised manuscript. The corresponding additions are provided below.

Page 13 (Discussion section)

This work builds upon our previous demonstration of striped ST-SPPs [42], sharing the femtosecond laser source and a single-nano-slit light-SPP coupling strategy. While retaining these foundations, the present study significantly extends the experimental framework to access a broader class of ST-SPPs [35]. Specifically, whereas the prior work treated separable ST-SPPs associated with a single transverse spatial frequency k_x , the current setup accommodates spatiotemporal spectra with finite spatial bandwidths, thereby enabling the excitation of non-separable ST-SPPs with tunable group velocity.

By incorporating an interferometric detection scheme, we can now obtain time-resolved measurements of the evolving ST-SPP field—rather than the time-averaged intensity profiles used previously. This capability allows direct evaluation of both the spatiotemporal structure of the wave packet and its propagation characteristics.

Since the reference pulse co-propagates with the SPP as shown in Fig. 3d, I would expect the beat and self interference frequencies to be different but they appear very close in the figure. Can the authors comment on this?

Authors' response:

We thank the reviewer for the constructive comment. In our experimental setup, both λ_{beat} and λ_{light} arise from the interference between the in-plane wavevector of the surface-propagating SPP and the in-plane projection of an obliquely incident optical pulse. Specifically, the interference pattern enclosed in the green circle results from right-propagating SPP ($+k_{\text{SPP}}$) and the reference pulse, whereas the interference pattern enclosed in the orange bounded area results from left-propagating SPP ($-k_{\text{SPP}}$) with the excitation pulse. The incident angle for the reference and excitation pulses are equal in magnitude but opposite in sign: $\psi_{\text{ex}} = -\psi_{\text{ref}}$. Therefore, beat wavelengths for two interference patterns are given by

$$\lambda_{\text{beat}} = \frac{2\pi}{|k_{\text{SPP}} - k_o \sin \psi_{\text{ref}}|}, \quad (\text{green circled area})$$

$$\lambda_{\text{light}} = \frac{2\pi}{|-k_{\text{SPP}} - k_o \sin \psi_{\text{ex}}|}. \quad (\text{orange circled area})$$

Consequently, $\lambda_{\text{beat}} = \lambda_{\text{light}}$. We have updated Fig. 3d in the revised manuscript to avoid any ambiguity.

The ST-SPP images recorded near the trench (Fig. 4) agree with the simulated field profiles. For verifying diffraction free propagation, it is not clear why 85 μm was chosen as the appropriate distance. There is a trade-off between dissipation and intensity, but farther away from the trench would make this result more convincing since the propagation length for a NIR SPP in Ag is at least twice that distance, in theory.

Authors' response:

We thank the reviewer for raising an excellent point. We limit our observation length to 60 μm in Fig. 4 as the axial length of the interference beat between ST-SPP and the reference pulse is approximately 30 μm for a fixed delay line position, as can be clearly observed in Fig. 6a. We do observe propagation of ST-SPP at the 100 μm distance from the slit by adding a delay in Fig. 5, second column.

Although the reviewer is right pointing out that near-infrared SPPs on silver can exhibit propagation lengths exceeding 100 μm under ideal conditions, in our experimental setup, a PMMA/dye overlayer was added to enable two-photon fluorescence imaging [41,55]. This overlayer alters the dielectric environment and increases modal losses, which in turn shortens the effective propagation length in our case. This reduction becomes particularly relevant in two-photon fluorescence microscopy, where the detected signal scales with the fourth power of the electric field amplitude. Therefore, we limited our analysis to within $\sim 100 \mu\text{m}$ from the excitation point, where both amplitude and phase information could still be reliably extracted.

In addition, the observation range in our experiment was primarily limited by the field of view of the microscope. At the magnification used to extract reliable phase information, the field of view was approximately $150 \times 150 \mu\text{m}$. To simultaneously capture both the slit and the center of the wavepacket within this area, the practical observation range was limited to around 100 μm .

The corresponding additions to the Method section are provided below.

Page 24 (SPP detection section)

In this study, we chose a field of view of approximately $150 \times 150 \mu\text{m}$, which allowed simultaneous imaging of both the light-SPP coupler and the full extent of the ST-SPP wave packet, while maintaining the spatial resolution required for accurate phase retrieval. As a result, the observation range was primarily limited to approximately 100 μm from the slit, despite the ST-SPPs propagating beyond this distance.

Page 24 (Calculating the SPP intensity profile section)

Therefore, the intensity, phase, and spatial distribution of the SPP wave packet can be extracted from the observed beat pattern through appropriate analysis. However, it is important to note that the beat pattern is not a direct representation of the SPP field itself. In particular, due to the interference-based nature of the detection method, the spatial frequency of the beat does not directly correspond to the SPP wavevector, and the detected signal decays more rapidly than the actual SPP field because it scales with the fourth power of the electric field amplitude.

A second point is that the profiles shown in Fig. 5 seem to be limited by the spot size of the reference beam. As such, if the ST-SPP diffracts beyond this area the signal will fade and give the appearance of diffraction free propagation. The concept of diffraction free propagation is supported by the theory provided, however, it is unclear to what extent the spatial convolution with the reference beam influences the Fig. 5 profiles.

Authors' response:

We appreciate the reviewer's constructive comment. We acknowledge that the finite size of the reference beam acts as a spatial detection window, potentially obscuring diffraction effects beyond the illuminated area. The reviewer is correct in pointing out this limitation of the current measurement scheme and that the theory predicts longer propagation distances. Due to the attenuation constraints described earlier, it was necessary to spatially confine the probe beam, particularly at locations farther from the light-SPP coupler, in order to maintain detectable signal levels in our current setup. However, by reference to Fig. 3f we show a conventional SPP of 5 μm width and observe clearly its diffraction, which proves that the finite size of the reference pulse does not represent an impediment to observe diffraction at 5 μm spatial scales.

We are currently working to expand the probe beam region in order to further extend the observable propagation range and to address the limitation of the current measurement scheme.

The ST-SPPs possess complex phase structure which could be exploited or may make some applications more difficult. Can the authors comment on how the phase profiles will influence some of the nanophotonic applications they propose such as surface-enhanced sensing?

Authors' response:

The reviewer is correct pointing out that ST-SPP in general have more sophisticated waveforms than conventional SPPs. This stems from non-separable spatiotemporal spectrum and intensity profile. Nevertheless, this does not necessarily entail that ST-SPP has a complex phase structure. Because each frequency component corresponds to a cosine wave [52], the phase alternates simply between 0 and π across the spectrum, rather than forming a more complicated phase structure. In fact, in our experiments we have maintained the phase flat across the entire spatiotemporal spectrum except for this 0– π alternation. We have corrected the wording in a few places in the revised manuscript (captions in the supplementary movies), which might have previously given the incorrect impression that the ST-SPPs possess complex phase structure beyond this simple alternation.

The reviewer nevertheless makes a good point, which is that the phase is an added degree of freedom that can be manipulated. For example, we have demonstrated previously that simply adding a π -step phase between the two halves of the spectrum results in a dark STWP in which a dip exists at the center rather than a peak. We expect that using such an STWP to excite an ST-SPP would result in a dark plasmon, in which a diffraction-free dip propagates in a metal—dielectric surface. Many applications can be envisioned for dark SPPs in addition to the ST-SPPs we have produced here, which we intend to investigate in the future.

Other comments:

- a) **Consider moving the last sentence of the Fig. 2 caption to the beginning of the caption to help guide the reader.**

Authors' response:

We agree with the reviewer, and we have implemented the suggested changes.

- b) **In Fig. 3b, the red excitation arrow is difficult to see since it aligns with the trench.**

Authors' response:

We agree with the reviewer, and we have implemented the suggested changes.

- c) **The text on page 10 refers to Fig. 3g but there is no such label for the field profile $U(x,z;t)$.**

Authors' response:

We thank the reviewer for catching this error. We have corrected it.

- d) **Consider making the labels c, d, g, and h bold in Fig. 4 like the others.**

Authors' response:

We would like to kindly mention that the labels are already in bold. However, we have moved them to the left for clarity.

We thank the reviewer for his very close reading our manuscript and providing thoughtful suggestions. We have revised the manuscript to implement the suggestions in figure captions, figure labels, and the text as indicated to improve clarity and consistency throughout the manuscript.

Responses to Reviewer 2

Reviewer's comment

The authors of the manuscript “observation of space-time surface plasmon polaritons” have experimentally prepared a non-diffracting space-time surface plasmon polariton (ST-SPP) wavepacket by coupling a precisely tailored optical field having a restricted 1D curvature in the spectral representation. Using two-photon fluorescence microscopy equipped with interferometry technique, they have unambiguously imaged the propagation of such ST-SPP on silver surfaces over tens of microns and hundreds of femtoseconds, and reconstructed its spatiotemporal field. Finally, the capability of tuning, despite the small range, of the group velocity of the ST-SPP wavepacket was shown. I find the result to be of interesting for the plasmon community, and the manuscript is well prepared.

Authors' response:

We thank the reviewer for taking time to review our work and the high evaluation of our manuscript.

I therefore suggest the authors to address the following points before I can recommend it for publication.

Q1. Since the excitation light for SPP fields needs spatiotemporal modulation of the wavefront, it unnecessarily inherits some spatiotemporal chirp of the excitation pulse. Could the authors discuss a bit of the effect of chirping, so that what would be the ultimate limit for the spatial confinement in the lateral dimension (k_z in Fig. 1) of the ST-SPP.

Authors' response:

From our previous experimental and theoretical research in STWPs in free space, there is no spatiotemporal chirp in the wavefront [28, 49]. Therefore, the ultimate limit of the spatial confinement along z is determined by the transform-limited pulse width associated with the spectral bandwidth effectively coupled to the SPP. In our experiment, however, there is some residual chirp that elongates the pulse from 10-fs to 16-fs stemming from the ultrashort nature of the laser source passing through various optical elements, such as lenses, gratings and SLM. Furthermore, any limiting of the bandwidth, either due to the nano-slit coupling or absorption at the silver-PMMA interface will further elongate the pulse and extend its length along z . In principle, this can be compensated through pre-compensation of dispersion using a chirped mirror or programmable phase control via the SLM. We have added the following sentence to clarify this point.

Page 22-23(Synthesizing free-space STWPs section)

Because of residual chirp ~~in the optical system~~ **accumulated through various optical components such as lenses, gratings, and the SLM**, the pulses are not transform-limited, and instead have a pulse duration of $\Delta T \approx 16$ fs at the sample surface.

Will the angle of excitation matter in generation of the striped ST-SPP?

Authors' response:

By reference to our previous theoretical analysis studying the coupling mechanism of ST-SPP [41] and previous work done by our group and others in ref. [54, R1, R2], the influence of the excitation angle on the generation of ST-SPPs is very small and can be considered negligible under our experimental conditions.

[R1] Buckanie, N. M.; Kirschbaum, P.; Sindermann, S.; Meyer zu Heringdorf, F.-J. Interaction of Light and Surface Plasmon Polaritons in Ag Islands Studied by Nonlinear Photoemission Microscopy. *Ultramicroscopy* 2013, 130, 49–53. <https://doi.org/10.1016/j.ultramic.2013.02.003>.

[R2] Wang, B.; Aigouy, L.; Bourhis, E.; Gierak, J.; Hugonin, J. P.; Lalanne, P. Efficient Generation of Surface Plasmon by Single-Nanoslit Illumination under Highly Oblique Incidence. *Appl. Phys. Lett.* 2009, 94, 011114. <https://doi.org/10.1063/1.3068747>.

Q2. The striped ST-SPP in Fig. 4c seem to have interference features extending upto 65 microns, much longer than that of the conventional SPPs in Fig. 4a, although they are measured at the same time, and their respective simulations have similar spatial distribution. So what is the cause of this “potential” elongation of the interference contrast?

Authors' response:

We appreciate the reviewer's comment. There are combination of two effects here.

First, each fluorescent image (Figs. 4a, 4c, 4f, and 4g) was acquired in a separate measurement. Since the fluorescent film experienced photobleaching during the time-resolved measurements, we had to use different regions of the sample to obtain the static images shown in these figures. As a result, slight variations in the experimental environment, such as local film thickness, led to small differences in effective group velocity for the measurements shown in Fig. 4.

Second, to capture a clear spatial distribution in Fig. 4, we used a relatively strong laser intensity and a long exposure time for each fluorescent image. Under these conditions, it was not feasible to acquire multiple images for time-origin correction, and the reference pulse delay was adjusted with limited timing resolution. The effective delay time in Fig. 4c is therefore estimated to be approximately 75 fs, with an uncertainty of about ± 5 fs.

These combined effects resulted in an apparent shift of the interference envelope toward longer z in Fig. 4c, giving the impression that the interference contrast extends farther than in Fig. 4a.

We note that the group velocity measurements were performed at a fixed position, with the laser intensity carefully adjusted to minimize photobleaching. In addition, the time origin was corrected by fitting the experimental data, allowing us to compensate for the timing offset. Therefore, the above variations do not affect the velocity analysis.

Page11 (Results section)

The differences in the wave packet center position and the extent of the interference envelopes are attributed to uncertainties in the time origin, spatial inhomogeneity in the fluorescent film, and variations in the intensity distribution of the probe beam.

Page12 (Results section)

All group velocity measurements were performed at a fixed location on the sample to eliminate variability arising from spatial inhomogeneity in the fluorescent film. Additionally, the delay time τ was adjusted such that the center of the ST-SPP wave packet appeared at $z = 0$ when $t = 0$.

Q3. Related to Q1, the lateral confinement (now becomes x-axis, however), is not much different in Fig. 4a and c. So what is the advantage of having striped ST-SPP in this case? I assume by having ST-SPP, one already reduces the intensity of the excitation light, and the coupling efficiency, possibly.

Authors' response:

Reviewer raises a crucial point. We have different spot sizes of the reference pulse in Fig. 4a-d compared to Fig. 4e-h. We listed them in the captions to the supplementary movies but did not list them in the caption to the Fig. 4, which we have corrected in the revised manuscript. The

spatial width along x in Fig. 4a and Fig. 4c are limited by the reference pulse spot size, which was $60 \times 15 \mu\text{m}^2$. Therefore in Fig. 4a and Fig. 4c the width along x was limited by the reference pulse although striped ST-SPP extends significantly beyond that, whereas the feature size of the striped ST-SPP is only $5 \mu\text{m}$. In the case of Fig. 4e and Fig. 4g for the subluminal and superluminal ST-SPPs we changed the reference pulse spot size to $30 \times 50 \mu\text{m}^2$. We extended the spot size along x in order to capture the tilt in the wavefront.

O4. The superluminal and subluminal propagation seem to have unsatisfying contrast in the experiments. Can the authors comment in a bit more detail why this occurs?

Authors' response:

We assume the reviewer is referring to plots in Fig. 5. Here we are plotting the time-averaged intensities of ST-SPP in contrast to the time-resolved measurements in Fig. 4, which have a better contrast. The reason is that in time-averaged measurements there is a background pedestal on the spatial profile that does not exist in time-resolved measurements; see ref. [28]. This background term does not exist for striped ST-SPPs because the spatiotemporal profile is separable. We have added a sentence clarifying this point on page 12 of the revised manuscript. The corresponding additions to the Discussion section are provided below.

Page 12 (Results section)

The reduced visibility in these time-averaged profiles compared to the time-resolved measurements in Fig. 4e,g is attributed to the integration over the full spatiotemporal field, which includes a broadband pedestal that partially obscures the localized structures.

This is rather critical in a sense that whether further tuning of the group velocity is feasible for ST-SPP.

Authors' response:

The pedestal height is fixed and independent of the group velocity for ST-SPPs.

For example, only limited tuning range was experimentally demonstrated, despite the theoretical curve has a broader capability of tuning.

Authors' response:

The reviewer is correct that pointing out the theoretical curve extends beyond our capabilities in tunability. In our experiment the limitation in the group velocity we can achieve is enforced by the numerical aperture ($\text{NA} \approx 0.09$) of the system and the spectral bandwidth exploited ($\approx 110 \text{ nm}$). The group velocity can be further tuned by either increasing NA or decreasing the bandwidth. As mentioned in the introduction we selected a large bandwidth and hence short pulse to resolve the propagation dynamics within the plasmon decay length.

If coupling efficiency reduces significantly for SPPs of different velocities, then it kind of loses the advantage as an information carrier.

Authors' response:

The coupling efficiency is independent of the group velocity. As mentioned above, the range of group velocity tunability is dependent not on the coupling efficiency, but the numerical aperture of the system and the spectral bandwidth of the source. We have added a comment on page 13 of the revised manuscript to clarify this issue. Indeed, as shown in our previous study [41] and a related conference publication [R3], we have confirmed through calculations that tuning the group velocity in fact does not affect the coupling efficiency as shown in the figure below (Fig. R1). To clarify this point, we have added the following text to page 13 of the revised manuscript:

Page 13 (Discussion section)

The demonstrated tuning range of $\tilde{\nu}$ was limited by the numerical aperture of our optical system ($NA \approx 0.09$), given the broad spectral bandwidth (~ 110 nm) required to ensure sufficient temporal resolution for resolving the propagation dynamics. However, the coupling efficiency is not significantly affected by the group velocity, and simulations have confirmed that ST-SPPs with a broader group velocity range can be excited [41]. Accordingly, broader tunability of $\tilde{\nu}$ could be achieved by increasing the NA or reducing the spectral bandwidth.

[REDACTED]

Fig. R1. ST-SPPs with a broad range of group velocities, as presented in Ref. [R3]. The superluminal case (1×10^9 m/s) and subluminal case (1×10^8 m/s) demonstrate that efficient excitation is maintained across widely varying group velocities.

[R3] N. Ichiji, M. Yessenov, K. L. Schepler, A. F. Abouraddy, and A. Kubo, "Propagation analysis of space-time surface plasmon polariton wave packet excited by light irradiation on nanobeam structure", Proceedings of SPIE 11999 1199904-1-1199904-7 (2022), <https://doi.org/10.1117/12.2607690>

We thank the reviewer for his very close reading our manuscript and providing suggestions. We have revised the manuscript to implement the suggestions in figure captions, figure labels, and the text as indicated to improve clarity and consistency throughout the manuscript.